# Controversies in the Surgical Treatment of Chronic Subdural Hematoma: A Systematic Scoping Review

**DOI:** 10.3390/diagnostics12092060

**Published:** 2022-08-25

**Authors:** Mary Solou, Ioannis Ydreos, Maria Gavra, Evangelos K. Papadopoulos, Stamatis Banos, Efstathios J. Boviatsis, Georgios Savvanis, Lampis C. Stavrinou

**Affiliations:** 12nd Department of Neurosurgery, “Attikon” University General Hospital, National and Kapodistrian University, Athens Medical School, 12462 Athens, Greece; 2Department of CT and MRI Imaging, “Agia Sofia” Hospital, 11527 Athens, Greece

**Keywords:** chronic subdural hematoma, treatment, surgery, burr hole, craniotomy, twist drill craniotomy, irrigation, drainage, MMA embolization, recommendations

## Abstract

Chronic subdural hematoma (cSDH) is one of the most common neurosurgical entities, especially in the elderly population. Diagnosis is usually established via a head computed tomography, while an increasing number of studies are investigating biomarkers to predict the natural history of cSDH, including progression and recurrence. Surgical evacuation remains the mainstay of treatment in the overwhelming majority of cases. Nevertheless, many controversies are associated with the nuances of surgical treatment. We performed a systematic review of the literature between 2010 and 2022, aiming to identify and address the issues in cSDH surgical management where consensus is lacking. The results show ambiguous data in regard to indication, the timing and type of surgery, the duration of drainage, concomitant membranectomy and the need for embolization of the middle meningeal artery. Other aspects of surgical treatment—such as the use of drainage and its location and number of burr holes—seem to have been adequately clarified: the drainage of hematoma is strongly recommended and the outcome is considered as independent of drainage location or the number of burr holes.

## 1. Introduction

Chronic subdural hematoma (cSDH), a collection of blood in the subdural space dated several days or weeks after the initial bleeding event, is one of the most common neurosurgical entities. The incidence is 1.72–20.60/100.000 persons annually, with morbidity and mortality rates of about 0–25% and 0–32%, respectively [1]. The majority of cases are recorded in the elderly population. With increasing life expectancy, the incidence of cSDH seems to be increasing as well [2]. Although cSDH is a treatable neurosurgical entity, the recurrence rate can be up to 28%, and up to 20% of patients with cSDH end up with a poor neurological outcome [1,2,3].

The clinical appearance of cSDH varies considerably: some patients are asymptomatic; some present with symptoms such as headache, walking instability, cognitive impairment or focal neurological deficits; while others have a severe clinical presentation, with stupor or even coma [4]. As symptoms are nonspecific, diagnosis is usually confirmed with non-contrast computed tomography (CT) of the head, or more rarely, magnetic resonance imaging (MRI) [4,5]. Research is focusing on several biomarkers such as inflammatory cell ratio, activated partial thromboplastin time and prothrombin time to assess cSDH severity and surgical outcome. Other biomarkers such as the brain natriuretic peptide have been investigated as prognostic markers for the long-term functional outcome of cSDH, while vascular endothelial growth factor (VEGF) concentration in cSDH has been investigated as a potential biomarker for hematoma recurrence [6,7,8]. Aside from CT and MRI, however, no other diagnostic tool has entered into daily clinical practice.

The management of cSDH may be conservative occasionally, but the surgical evacuation of the hematoma remains the mainstay of treatment. A large number of studies examine the optimal therapeutic management of cSDH in regard to neurological outcome, complications and recurrence rate. Still, many controversies surround the surgical therapy of chronic subdural hematoma that restrict the formation of specific guidelines, or at least a general consensus among neurosurgeons. Addressing these issues where consensus is lacking and updating our knowledge on them is important, as it will help to improve the outcome of patients with cSDH, but also highlight the areas where further research is needed.

## 2. Materials and Methods

To address the controversies with respect to the surgical treatment of chronic subdural hematomas in the contemporary era, a systematic review was performed. This manuscript was structured in accordance with “The Preferred Reporting Items for Systematic Reviews and Meta-Analysis (PRISMA) guidelines” [9].

Information Sources and Search Strategy:

Articles published between January 2010 and May 2022 regarding the surgical management of chronic subdural hematomas were searched using the Pubmed search engine. The following algorithm was used: “Chronic subdural hematoma AND (treatment OR management OR therapy OR intervention OR surg* OR surgical indications OR timing for surgery OR complications OR recurrence OR twist drill craniotomy OR burr-hole craniotomy OR craniotomy OR mini-craniotomy OR endoscop* OR irrigation OR drainage OR membranectomy OR (embolization AND middle meningeal artery))”. The reference lists of the selected articles were also reviewed.

Eligibility Criteria:

Included were studies written in English that had an abstract and full text available, a specific study design (meta-analysis, systematic review, randomized controlled study, prospective and retrospective cohort study, case series and review-type study), and referring to a population of adults (defined as age ≥18 years old) with a diagnosis of cSDH who were treated with a type of surgical intervention for the evacuation of the hematoma, or reviewing aspects of the surgical treatment of cSDH. Excluded studies were those with only an abstract available, case reports, children population studies, papers published in languages other than English and those published prior to 2010.

Selection and Data Collection Process:

The selection of study items followed a double-reviewer process (L.C.S., Prof. of Neurosurgery and M.S., Senior Resident in Neurosurgery). The records identified in PubMed using date filters (01/2010–05/2022) and the foretold keywords were checked for duplicates using Mendeley reference management software. Then, articles were screened according to the selection criteria mentioned above. The retrieved papers were examined in full for eligibility. In the final list of articles included in this systematic review, some studies from the respective reference lists were included. After articles were selected for inclusion, we reviewed the characteristics of each study, including design, aim, surgical intervention(s)/ method(s) used and outcomes. Data were divided in categories (each representing a step of the surgical management of cSDH) for further analysis. Subdivisions included indications of surgery, timing of surgery, surgical method, membranectomy, the number of burr holes, irrigation, drainage (implementation, localization and duration), and embolization of the middle meningeal artery (MMA).

Data Items and Outcomes:

Recurrence rates and need for re-operation were considered primary outcomes. Secondary outcomes included morbidity, mortality and postoperative complications. All the included studies were assessed for the level of evidence and quality, then, conflicts were discussed and resolved.

Study Synthesis:

The systematic review was structured by analyzing the data from each separate category. For examining the controversies, all the ambiguous data were demonstrated analytically by authors in text-form or table-form. Other tables were created to illustrate more specifically the results of this systematic review. Microsoft^®^ Excel was used for chart making.

## 3. Results

The search identified 3492 studies on PubMed. After screening the records on a title–abstract basis, 2832 studies were excluded due to inconsistent research content. Apart from 310 reports that could not be retrieved, 350 full-text studies were assessed for eligibility, and after a double-reviewer process 254 records were removed. Therefore, 96 studies were identified as eligible and 15 additional papers were added to this review from the reference lists. In total, this systematic review included 111 studies (Figure 1).

### 3.1. Data of Included Studies

In this systematic review, data were derived from N studies of different types (meta-analysis, systematic review, randomized controlled study, clinical study, case series and review-type study), as illustrated in Figure 2. The possible controversies are shown in Table 1 and constitute the categories in which data from the included studies were divided for analysis.

### 3.2. Result Analysis

Table 2 presents the main controversies and the respective recommendations emerging from the analysis of each separate aspect of the surgical management of cSDH. The grade of recommendation (A, B or C) is based on the evidence of the analyzed studies (Appendix A).

## 4. Discussion

While the diagnosis of chronic subdural hematoma is easily established through a head CT scan, an increasing number of studies are focusing on biomarkers in an attempt to reveal a relation to optimal treatment, recurrence and outcome. Pripp et al. in their cohort study analyzing 30 biomarkers of inflammation and angiogenesis, and several imaging characteristics in cSDH patients, concluded that the density and volume of cSDH pre- and postoperatively, and biomarkers such as pro-inflammatory cytokine CXCL8 and possibly interleukin Il-6, may be predictors of post-operative recurrence, even if they are not related to the clinical status of cSDH patients [12]. Other studies are mostly focused on the association of biomarkers and imaging signs with the final outcome. A study by Idowu et al. showed that the initial values of activated partial thromboplastin time (APTT); prothrombin time (PT); international normalized ratio (INR); and platelet-to-lymphocyte ratio (PLR) are good prognostic factors when using the Glasgow outcome scale (GOS) to measure the outcome [8]. They also demonstrated that a high international normalized ratio (INR) at patient’s admission is a negative prognostic factor for the outcome based on the Markwalder grading scale (MGS) and Lagos brain disability examination scale (LABDES) [8]. However, no studies have resulted in a correlation between biomarkers and the optimal surgical therapy of cSDH.

### 4.1. Indications of Surgery

The decision to surgically intervene in cSDH largely depends on its clinical presentation and radiological characteristics, such as hematoma size, midline shift, the presence of membranes and the presence of bilateral hematomas [13,14]. It is generally accepted that patients with neurologic symptoms and relevant radiological findings should undergo surgical evacuation. On the other hand, asymptomatic patients with no compression signs in imaging are usually managed conservatively [13,15,16,17,89,90]. A surgical approach is also advised when the neurological status deteriorates, even without worsening radiological findings. Aside from the above fairly straightforward scenarios, there is still much controversy regarding the decision to surgically treat a cSDH. The optimal treatment of asymptomatic patients with radiological findings of brain compression and midline shift, and of symptomatic patients with minor radiological findings, remains a matter of debate. There are no studies available comparing surgical vs conservative management in this group of patients. There is, however, clinical consensus that hematomas of a thickness greater than 1 cm, or equal to or exceeding the thickness of the skull, should be evacuated. Similarly, a midline shift of more than 5 to 7 mm warrants surgery [15,16,17,18,19,20,91]. Sahyouni et al. reported a cut-off value of 7–10 mm for midline shift, which seems to arise from the literature for acute subdural hematomas, and is, therefore, debatable whether it applies to chronic subdural hematomas as well [13]. To date, no clear cut-off values exist for midline shift and in this regard the decision to operate remains largely empirical [15]. Other radiological findings can also be taken into consideration when deciding on the best treatment for this group of patients, such as the effacement of basal cisterns or the presence of brain atrophy. However, how much each of these findings weighs in the treatment decision is not clear. Again, in regard to the second group of patients—namely with symptoms but minor radiological findings in the absence of evidence-based data—the consensus is that once other causes have been excluded (for example stroke), surgery is advised.

The spontaneous resolution of cSDH does occur, but no standard clinical or imaging signs have been proposed to predict whether a cSDH will resolve spontaneously [16]. Soleman et al. referred that the autonomous resolution of thick cSDH has so far been reported in one study examining elderly patients with brain atrophy and absent signs of elevated ICP, although in the same article, the authors highlighted that comparative studies between conservative and surgical management in this group of patients are lacking [15]. In general, conservative treatment is commonly reserved for patients with minor symptoms, for example, Markwalder score 0–1, individuals with considerable operative risk and for those denying any surgical intervention [16,18,89,92,93].

The term cSDH typically refers to supratentorial hematomas. Chronic subdural hematomas of the posterior fossa do occur, but they are rare; in adults, they are the result of trauma or other spontaneous causes [21,22,94,95]. Apart from case reports and small case series, the literature is scarce, making the formulation of treatment recommendations difficult. Although a few reports on the conservative management of cSDH in the posterior fossa do exist, the majority of these hematomas have been treated surgically with either craniotomy, craniectomy or burr holes [22,94].

### 4.2. Timing of Surgery

While surgical time frames have been extensively studied in acute subdural hematomas, these do not pertain to cSDH [23]. Therefore, clinical practice is mostly based on the reasonable consensus that cSDHs should be ideally treated as soon as possible, especially in patients with neurological deficits. While most papers highlight the fact that cSDH should be treated “timely”, only a few studies have studied this variable in chronic subdural hematoma treatment. Venturini et al. in their 2019 observational cohort study examining 656 patients with cSDH concluded that the time of surgery is associated positively with the length of hospitalization, but not with the outcome, complications, recurrence, reoperation and survival [23]. Additionally, they showed that a time of surgery later than 7 days after symptoms begin decreases the chance of a favorable outcome at discharge. Most patients underwent surgery on the first day, which illustrates the timely manner in which they were treated [23]. On the other hand, Zolfaghari et al. were not able to identify any significant negative effect upon outcome correlated to time from diagnostic CT scan until surgery. It should be noted, however, that most patients were in good condition (GCS >13) preoperatively and that the mean time to operation was 76 h [24]. Sometimes, there are practical reasons for which surgery should be delayed, even in symptomatic patients. Chronic SDH affects mostly elderly people who are often treated with anticoagulants. It is generally agreed to reverse coagulopathy before cSDH surgery is undergone [15]. Based on the new European Heart Rhythm Association (EHRA) guidelines and the British Society of Haematology guidelines, a 24-h delay of surgery is advised in patients systematically treated with new oral anticoagulants (NOACs), counting from the last dose taken. The delay may be extended to 48 h when renal impairment or direct thrombin inhibitor are part of a patient’s medical history [17,23,25]. If urgent surgery is required, guidelines propose prothrombin complex concentrate (PCC) for reversing the effects of NOACs [25]. As far as antiplatelets are concerned, the delay of operation should be extended to 7–10 days while waiting for platelet renewal. In life-threatening situations, the American Society of Hematology guidelines suggest platelet transfusion before surgical intervention [17].

### 4.3. Surgical Method

The surgical treatment of cSDH is primarily based on one of the following techniques: twist drill craniostomy (TDC); burr hole craniostomy (BHC); craniotomy; and endoscopic evacuation. Although BHC is the treatment of choice for cSDH in most neurosurgical departments and is performed frequently, many controversies and questions concerning the operational techniques remain unanswered. Given the frequency by which neurosurgeons are confronted with cSDH, the amount of class I evidence regarding the preferrable surgical treatment is astonishingly small.

#### 4.3.1. Twist Drill Craniostomy (TDC)

Historically, TDC with a closed system drainage was firstly reported by Tabadoor and Shulman in 1977 in their cohort of 21 patients treated for cSDH [17,89]. It is a minimally invasive procedure involving the creation of a small opening to the skull with a diameter of about 2–5 mm (generally less than 10 mm), usually using handheld drills [15,17,18,89,90]. It has the advantage that it can be performed at the bedside under local anesthesia and can, therefore, be a very attractive management option for polymorbid cSDH patients who are poor surgical candidates [15,17,18,26,89,90]. It is reported that by using TDC with a closed drainage system for continuous drainage, surgeons can achieve slower brain decompression, better re-expansion and the avoidance of complications that may arise due to rapid evacuation, such as intraparenchymal hemorrhage [15,89,90]. Intraparenchymal hemorrhage or seizure after the rapid decompression of cSDH may happen in 60% of patients aged above 75 years, possibly owing to the excessive hyperemia in the healthy cortex beneath the hematoma [27,89,90]. The intraoperative use of irrigation is still a matter of debate. Some of the disadvantages include inadequate drainage, brain penetration, acute epidural hematoma, catheter folding and contamination risk in cases of bedside TDC [15,18]. Modified TDC techniques have been implemented to address these issues, such as using a hollow screw placed into the skull to set up a closed drainage system, thus, helping to minimize the risk of complications due to blind catheter insertion. However, more research needs to be carried out to determine the efficacy of such techniques [17,89,90]. In general, TDC is regarded to be mostly effective when hematoma is completely liquified with minimal membrane formation [15].

#### 4.3.2. Burr Hole Craniostomy (BHC)

First described by Svien and Gelety in 1964, BHC is a minimally invasive technique which involves the drilling of one or two burr holes of 10–20 mm in diameter [13,14,17,18,89,90]. If two burr holes are drilled, they usually are placed 5–8 cm apart [17]. Following this, the subdural hematoma is irrigated with saline until the fluid runs clear [17,89]. Some surgeons utilize silicone soft drain to complete the evacuation of the hematoma intraoperatively, which can also be used as closed system drainage up to 48 h postoperatively [17,89]. BHC can be performed either using general anesthesia or conscious sedation based on the surgeon’s preference, patient’s tolerance, compliance, and comorbidities, although there are reports of BHC being performed at the bedside under local anesthesia [13,20,89]. Although BHC is performed very frequently all around the world, it is surrounded by a cloud of controversies: The number of burr holes (one vs two) and their location, the use of intraoperative irrigation and the location of drain placement are all addressed in the literature [90].

#### 4.3.3. Craniotomy

Until approximately the 1970s, craniotomy was traditionally the treatment of choice for cSDH. With the advent of routine CT imaging, its use has decreased significantly and is nowadays mostly reserved for recurrent cases of cSDH or hematomas with extensive membrane formation or calcifications [15,17,18,89]. A craniotomy is the creation of a larger free bone flap—usually more than 30 mm—to expose the greatest portion of the subdural hematoma covering the brain [15,17,89,90]. Despite its benefit of extensive evacuation of the hematoma and maximal access for excision of membranes, it is the most invasive of the techniques and it is associated with a longer operating time, larger amount of blood loss, more post-operative complications and longer hospitalization times, especially in frail patients [15,28,89]. In a meta-analysis by Lega et al. craniotomy yielded fewer recurrences, but had a greater complication rate [29]. Routinely performing a membranectomy following craniotomy for cSDH is a matter of controversy [89]. In recent years, extensive craniotomies have given their place to smaller mini-craniotomies, which appear equal to BHC and TDC in regard to invasiveness and complication rate, but with superior visualization and lower recurrences [30].

#### 4.3.4. Endoscopic Procedures

In recent years, the endoscope-assisted evacuation of cSDH has gained traction. A mini-craniotomy or an enlarged burr hole is drilled in the skull, from which the endoscope is inserted [31]. Using direct visualization via endoscope, the drainage of cSDH becomes safer and more effective. A prospective study of 72 hematomas managed endoscopically showed that thick, vascularized membranes, septations and solid clots can be removed easily using this technique [18,31]. However, more studies are needed to determine the limitations of endoscopic procedures.

### 4.4. Comparison of Surgical Techniques

Controversy exists in almost every aspect of the surgical management of cSDH, from the type of surgical technique to the number of burr holes, their location and the use of irrigation, to name a few. Unfortunately, evidence is mostly based on meta-analyses and single-center retrospective studies (class II and III evidence), thus, making the formulation of recommendations challenging.

#### 4.4.1. TDC vs. BHC vs. Craniotomy

Only a few studies compare all three of the most commonly used surgical techniques and none of them offers class I evidence. A 2003 metanalysis by Weigel et al. supports that TDC and BHC have a better safety profile when compared to craniotomy, for which morbidity rates were significantly higher at about 12.3% [10]. Differences in cure rates did not reach statistical significance. Both burr hole craniostomy and craniotomy had lower recurrence rates than twist drill craniostomy [10]. Additionally, in recurrent cSDH, it seems that BHC is more effective than TDC or craniotomy, which should be considered the last-choice treatment [10]. The authors concluded that twist drill and burr hole craniostomy can be considered first-tier treatment, while craniotomy may be used as second-tier treatment. Cofano et al. examined recurrence rates after surgery in a 2020 multicenter cohort study and concluded that burr hole craniostomy is associated with lower recurrence rates, when compared to operative methods (9.3 vs. 18.8%, respectively) [32]. A 2022 RCT comparing BHC, mini-craniotomy and TDC concluded that all three techniques are effective in treating patients, with 6-month outcomes being similar [33]. BHC appeared to offer the lowest recurrence rate at a manageable complication rate, although this difference did not reach statistical significance [33]. Ducruet et al., in their meta-analysis of 2012, recommended that TDC with drainage at the bedside should be the primary treatment choice for high-risk surgical candidates with non-septated cSDH, while craniotomy should be selected for cSDH with multiple membranes [34]. TDC appeared to produce the best outcome and the fewest complications rate compared to BHC and craniotomy, while mortality rates appeared to be higher in cases where craniotomy was performed. Another meta-analysis of 34829 patients by Almenawer et al. found no significant difference among the various surgical techniques regarding morbidity, mortality, outcome and recurrence rates [26]. The authors report that craniotomy is more efficient in cases of recurrent cSDH, but is also associated with the greatest complication rate [26]. Lega et al., in their analysis using a multiple probability simulation, concluded that BHC balances the lowest rates of recurrency and complications, and thus, is the overall most efficient choice [29]. However, the lowest recurrence rate, but also the highest complication rate, belonged to patients treated with craniotomy [29].

#### 4.4.2. BHC vs. Craniotomy

A number of studies focus on comparing BHC to craniotomy: Mondorf et al. conducted a retrospective study examining the outcome and recurrence of 193 patients with cSDH treated with craniotomy (151 patients) or BHC (42 patients) [96]. Their results showed that recurrence happened in 27.8% of patients in the craniotomy group and 14,3% in the BHC group patients. In the same study, about 52.3% of patients treated with craniotomy had complete neurologic recovery at discharge, while the respective percentage in the BHC group was 64.3% [96]. The authors concluded that burr hole drainage is better in terms of recurrence rate and the recovery of symptoms than craniotomy [96]. In a retrospective study by Shim et al., the authors examined the recurrence and the duration of the hospitalization of patients treated for cSDH, either via BHC or a small craniotomy [35]. Their findings support the notion that BHC is superior to a small craniotomy, as it had a lower recurrence rate (13.3% in BHC Vs 26.7% in small craniotomy) and a shorter average hospitalization time (10.3 days for the BHC group vs 15.7 days for the craniotomy group) [35]. However, this study seems to suffer from an intrinsic bias as to the technique used, because cSDH with septi tended to receive a small craniotomy [35]. A small single-center 2020 study presented different findings, as its results showed less recurrence in patients treated with craniotomy when compared with those who underwent BHC [36]. Moreover, a single-center retrospective analysis by Gazzeri et al. comparing four groups of patients with cSDH treated with BHC or craniotomy using either subdural or subgaleal drainage, found the recurrence rate and neurological outcome to be independent of these two surgical techniques and the drainage location and, thus, the authors suggest a personalized selection of the technique for the treatment of cSDH [37].

#### 4.4.3. TDC vs. BHC

The only RCT comparing the two minimal procedures for the management of cSDH is a study by XU et al., which showed no significant differences between the cure and mortality rates of patients treated for cSDH [38]. In regard to neurological outcome, however, TDC appeared superior to BHC, as the mRS score at the 3-months follow-up was significantly improved in the TDC group compared with that in the BHC group, and the overall length of hospitalization was significantly shorter when TDC was performed [38]. The clinical equipose between twist drill craniostomy and burr hole craniostomy was addressed in a meta-analysis by Yagnik et al. in 2021 [2]. They performed a systematic review and meta-analysis comparing outcomes following BHD and TDC for initial surgical management in cSDH. Although complications, recurrence, cure and mortality rates were not significantly different between the two, TDC was associated with a higher reoperations rate than BHD [2].

#### 4.4.4. Endoscope-Assisted BHC (EBHC) vs. BHC/Craniotomy

The newest technique for treating cSDH, the endoscope-assisted evacuation of cSDH, is compared to BHC in a meta-analysis conducted by Guo et al. [40]. Their results showed that recurrence rate and complications were significantly decreased in the group of patients treated with endoscope-assisted surgery [40]. A 2018 retrospective study by Zhang et el. compared endoscope-assisted burr hole craniostomy to ordinary BHC and found the endoscope-assisted technique to be superior in lowering recurrence rate, morbidity rate, duration of drainage and length of hospital stay [39]. Similarly, a retrospective comparative study between 97 endoscopically treated patients and 380 patients treated with a classic BHC found a lower rebleeding and re-operation rate in favor of the endoscopic technique [41]. The advantage of EBHC remained even when the analysis included only complicated cases, i.e., those with the presence of clot and/or septi [41]. EBHC appears advantageous when compared to craniotomy, too. In a retrospective study by Zhang et al., the endoscopy group had less blood loss and shorter hospital stays [42]. These findings, however, are not uniform in all studies; a retrospective analysis by Yan et al. found no difference between BHC and EBHC with respect to the hematoma recurrence rate (8.7% and 13.7%, respectively). The authors concluded that in light of this finding, BHC appears as the better choice, as it requires less surgical time [43].

#### 4.4.5. Refractory cSDH

The recurrence of cSDH after surgical treatment remains a major issue, with 5% to 10% of patients requiring repeated operation after 30 to 90 days [44]. In particular, refractory cases, sometimes defined as more than two recurrences, pose a special challenge. Matsumoto et al., in their cohort study examining refractory cases of cSDH, showed no significant difference in cure rate between patients treated with burr hole irrigation and drainage alone, and patients treated with burr hole irrigation and drainage with embolization of the MMA. Similarly, no significant differences in cure rate were seen between patients treated with burr hole irrigation and drainage alone and patients treated with craniotomy. It should be noted, however, that the lack of statistical significance between the groups could be due to the inadequate powering of the study (14 patients) [45].

### 4.5. Number of Burr Holes

The number of burr holes (one or two) might seem trivial; however, performing two burr holes means a longer operation and two wounds for post-operative care. In that regard, the question is whether a single burr hole is sufficient to evacuate the hematoma and whether the recurrence rate is related to the number of burr holes. Mersha et al. confirmed the safety and effectiveness of a single burr hole in a retrospective study of nearly 200 patients [46]. At the same time, Abdelfatah et al. examined 47 patients who underwent a double burr hole evacuation for cSDH and found that two burr holes evacuate the cSDH efficiently, without reporting any recurrence [97]. Most of the comparative published studies support the equality between single and double burr holes in cure and recurrence rates. To confirm this, in 2021, Sale et al. performed an RCT studying 192 patients [98]. They reported that one burr hole has a similar outcome and recurrence rate to double burr holes whilst having shorter operative time [99]. Similar results have been recorded by Nayil et al. in their RCT of 258 patients in 2014 [47]. The fact that the number of burr holes does not affect the recurrence rate when comparing single versus double burr holes has been reported by numerous cohort studies [48,49,50,99]. Some of them are a part of meta-analyses by Almenawer et al. in 2014, Belkhair et al. in 2013, Smith et al. in 2012, and Wan et al. in 2019, which conclude that there is no significant difference between one and two burr holes in terms of recurrence risk [26,51,52,53]. Nevertheless, the literature is not unanimous; Han et al. and Khan et al., in their respective cohort studies, support the notion that a single burr hole is better than the double burr hole, which resulted in a higher recurrence risk [54,55]. In contrast, an older study by Taussky et al. in 2008 reported that the use of a single burr hole is associated with higher recurrence rates, longer hospitalization and a greater wound-infection risk [56]. Based on the aforementioned RCTs and meta-analyses, it seems that there is no discernible difference between one and two burr holes and that the choice is a matter of the surgeon´s preference.

### 4.6. Irrigation

Intraoperative irrigation theoretically clears the hematoma elements and reduces the risk of recurrence. The literature, however, seems to be more nuanced. Some authors support the notion that irrigation after BHC is related with lower recurrence rates when compared to drainage alone [10,18,100]. Two older studies have reported that inflow and outflow irrigation in BHC is associated with fewer recurrence rates, and one study came to a similar conclusion when a TDC was used [101,102,103]. Most authors, however, argue that the outcome with or without irrigation is the same in cSDH managed by a drainage system [18,29,57,58,104,105]. This is confirmed by meta-analyses conducted by Xu et al. and Yuan et al. [59,106]. A retrospective study including 385 cases of cSDH that were managed with BHC drainage without irrigation showed that the recurrence rate was only 4.9%, supporting the idea that irrigation may not be necessary for every patient [107]. Another study conducted in two medical centers in China concluded that irrigation in BHC for the management of cSDH does not offer any further curative outcome and is closely related with short-term complications such as pneumocephalus [108]. When examining the use of intraoperative irrigation, further questions arise as to whether it should be combined with aspiration, what type of solution should be used, and whether it should be applied via a catheter or directly through the burr hole. A small study of 51 patients showed that combining irrigation with aspiration results in a significantly lower recurrence rate and a better outcome when compared to irrigation alone [109]. Regarding the type of solution used for irrigation, Ivamoto et al. found that the irrigation of cSDH with thrombin solution may decrease this possibility of recurrence in patients with a high risk of recurrence [91]. A small retrospective study examining the temperature of irrigated fluids demonstrated that fewer recurrences occur when the irrigation fluid is at body temperature instead of room temperature [60]. Kuwabara et al. studied the outcome and recurrence rates when managing cSDH with irrigation, using either artificial cerebrospinal fluid or normal saline, and showed that artificial cerebrospinal fluid is associated with decreased recurrence rates [61]. Lastly, a retrospective cohort study comparing the irrigation via a catheter versus a siphon irrigation—where fluid is irrigated directly through the burr hole and drained through an in situ drainage—showed that the siphon group patients had a better evacuation in the postoperative CT image, earlier neurological recovery and less days of hospitalization [62]. For now, it seems safe to say that irrigation may be omitted when a closed-system drainage is used.

### 4.7. Closed System Drainage

The use of closed-system drainage is one of the few parameters in the surgical management of cSDH for which there is a type A recommendation [15,63]. An important milestone in the literature of cSDH was an RCT by Santarius et al., which showed a benefit in recurrence, mortality and hospitalization days after subdural drain placement following BHC for the evacuation of cSDH [15,17,63]. Since then, several studies have shown that the placement of a continuous drainage system after cSDH evacuation aids in clearing the residual subdural fluid, which contains inflammatory and fibrinolytic factors and is related to early neurological improvement and a decrease in recurrence rates, with Peng et al. reporting up to 50% minimization of the recurrence risk [64,65,89]. Studies exploring the use of external drainage in cSDH therapy unequivocally support its beneficial role regarding recurrence rates [18,63,66,67,68,91,110]. Two meta-analyses and a Cohrane review of RCTs by Alcala-Cerra et al. confirmed the conclusion that the use of postoperative drainage in cSDH reduces the recurrence risk and improves outcome, without additional complications [26,64,65]. Moreover, continuous drainage for cSDH treatment is associated with less hospitalization days and less risk for pneumocephalus [18,68]. A more recent RCT also showed a decrease in recurrence risk from 24.3% to 9.3% and a reduction in mortality rates at 6 months from 18.1% to 9.6% when a subdural drain was used [63]. In conclusion, there is strong and increasing evidence that the placement of a continuous drainage system after CSD evacuation is advantageous (grade A recommendation).

#### Drainage Location

Even though the insertion of a subdural drainage system is considered safe and efficacious, its use is not without complications. These may include acute hemorrhage from neomembranes, parenchymal injury, tension pneumocephalus, meningitis, subdural empyema, complications related to prolonged immobility, and seizures [15,18,64,65,89]. Consequently, some authors advocate a subperiosteal drainage placement as being less invasive, with a potentially better mortality and complications rate, whilst being as efficacious as a subdural placement [18,69,70,71,72,73]. A 2021 RCT with 42 patients found that that the drain type (subdural or subperiosteal) has no effect on the outcome [74]. A number of studies found similar rates of recurrence, regardless of drainage location or the use of anticoagulants [15,71,75]. Gazzeri et al. found no significant difference in recurrence rates and functional outcomes, comparing the use of the subgaleal vs subdural location of drainage [37]. However, the majority of authors indicate that subperiosteal drains combine equal recurrence but lower infection and drain misplacement rates compared to subdural drains [90,111,112,113]. Greuter et al. found that while recurrence rate does not significantly differ between subperiosteal and subdural drains, drain misplacement is higher in subdural groups [70]. Therefore, they recommend the placement of a subperiosteal drain in the elderly (over 80 years old), or in patients at a high risk for complications (Grade B recommendation) [15,69,70]. The cSDH-Drain-Trial was a multicenter, prospective, randomized, controlled trial analyzing the recurrence indicating a re-operation in adult patients undergoing burr hole drainage (subdural or subperiosteal) [75]. Their results showed that even the noninferiority criteria of 3,5% were not met, and the subperiosteal insertion of drain had less recurrence rates, fewer surgical infections, and lower drain misplacement rates [75]. On a similar note, a 2019 multicenter cohort study of 570 cases showed that the outcomes of subdural and subperiosteal drains after burr hole craniostomy for CSDH are largely equivalent [76]. A 2020 meta-analysis largely confirmed the abovementioned findings, i.e., subperiosteal drainage placement is as effective and at least as safe as subdural placement [112]. In conclusion, subperiosteal or subgaleal drainage placement appears to be as effective as subdural placement with comparable recurrence rates and a potentially lower complications rate (Grade A recommendation).

### 4.8. Duration of Drainage

The optimal duration of drainage after surgery is yet another controversial issue in the management of cSDH. A small study investigating the duration of subdural drain in cSDH treated with BHC demonstrated that the time of drainage does not significantly affect the recurrence risk or the possibility of infection [99]. Similar results were revealed by a national Danish randomized clinical study regarding 24-h versus 48-h postoperative subdural drainage in patients who underwent a single burr hole for cSDH [77]. A subgroup analysis of the UK Multicenter Prospective Cohort Study found comparable recurrence rates between patients with postoperative drainage of one or two days (6,4 vs 8,4%, respectively) [78]. However, a 2017 retrospective study examining the drainage duration in a total of 90 patients concluded that closed system drainage for 2–4 days following burr hole craniotomy can be an effective choice, but it is related with a higher risk of recurrence compared to preserving the drainage for 5–7 days [79].

### 4.9. Membranectomy

Historically, performing craniotomy in patients with cSDH was combined with a membranectomy, as it was believed to be necessary for a successful hematoma evacuation and brain re-expansion [17]. Indeed, the results from some contemporary cohort studies suggest the advantage of the membranectomy technique: A study by Kim et al. compared mini-craniotomy with partial membranectomy to large craniotomy with the extended excision of membranes and concluded that recurrence rates were significantly lower in the second group [80]. Another study by Elayouty et al. was based on performing BHC with restricted membranectomy. The authors reported that adding membranectomy to BHC reduces the risk of recurrence [81]. Sahyouni et al. conducted a meta-analysis of 17 cohort studies, showing that craniotomy with the excision of membranes is related to a lower recurrence risk but similar mortality and morbidity rates when compared to the literature rates for craniotomies without membranectomy [82]. However, there are some studies resulting in no significant statistical difference in re-operation rates, regardless of whether membranectomy was performed or not [83,84].

### 4.10. Embolization of Middle Meningeal Artery (MMA)

Middle meningeal artery embolization aims to reduce the vascular products that reinforce the enlargement of subdural hematoma and the subsequent creation of neomembranes from vascular debris [20,45,90]. In a retrospective study of 55 patients with cSDH, Takizawa et al. reported a larger diameter of MMA in magnetic resonance angiography in all patients. This information may be significant in developing a strategy for the treatment of cSDH [85]. The general consensus of 12 RCTs, two non-randomized controlled studies, two prospective single arm trials, one combined prospective and retrospective controlled study, and one prospective cohort study is that MMA embolization, when applied as a primary standalone treatment, in recurrent cSDH, or as a prophylaxis after surgery, is a safe and efficient method, with lower recurrence rates when compared to the conventional management of cSDH [86]. There is, however, significant heterogeneity in data element collection, making the formulation of type A recommendations premature [3,86,87,88,114,115,116,117,118,119,120,121,122,123]. In four of the most recent meta-analyses by Ironside et al. (2021), Srivatsan et al. (2019), Haldrup et al. (2020) and Jumah et al. (2020), the recurrence rates in patients with cSDH who underwent the embolization of MMA either as a primary treatment or in recurrent cSDH were 4.8%, 2.1–3.6%, 2.4–4.1% and 2.8%, respectively [114,120,121,122]. These results were lower when compared to those of conventional treatments. MMA embolization in these studies was not associated with any difference in complication rates. The embolization of MMA is increasingly becoming an established technique to treat cSDH, especially in refractory cases. In 2022, Nia et al. examined the use of MMA embolization as a primary standalone method for managing cSDH. They showed favorable treatment failure rates when compared to surgery for a specific patient population: their indications for the primary use of MMA embolization were of aged people (72 ± 12 years old) with a substantially high comorbidity index of 7,02 [124]. It is, however, a method that is not applicable in every patient and every institution, and the eligibility criteria are yet unclear [20,45]. Severe renal failure or no available access routes to MMA are classic contraindications [45].

## 5. Conclusions

Chronic SDH represents one of the most frequent neurosurgical entities, especially in elderly patients. With an ever-aging population, an increase in incidence rate is expected. Many aspects in the management of cSDH remain controversial. Given the frequency of the disease, the number of well-designed randomized trials generating high-level therapeutic recommendations is surprisingly small. It is generally accepted that in the presence of neurologic symptoms and radiologic findings, patients should undergo surgical evacuation. Burr hole evacuation (one or two) seems to be the preferred surgical method, as it offers the best cure-to-complications ratio, followed by subdural or subgaleal drainage placement for 24 or 48 h. The efficacy of hematoma irrigation and the need for membranectomy are still insufficiently clarified. Prospective multicenter studies providing type A recommendations for these questions are much needed.

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
