# Peer review of "Controversies in the Surgical Treatment of Chronic Subdural Hematoma: A Systematic Scoping Review"

_diagnostics, 2022, doi:10.3390/diagnostics12092060_

Round 1

Reviewer 1 Report

Title: Controversies in the Surgical Treatment of Chronic Subdural Hematoma: A Systematic Scoping Review

The authors reviewed literatures of CSDH cases which were published from 2010 to 2022 and discussed the controversies in the Surgical Treatments.

Recently, with the increasd data accumulation in PubMed, many poorly written papers that only summarize past literatures are increasing. A systemic review is OK. However, the authors should analyze their own data first and compare them with the past huge data analysis to discuss the authors originality, and scientists should not write papers using only data that others have painstakingly built up. If this kind of paper can be easily accepted, even if it is a clinical paper, the authors can write the paper without contacting patients, without performing surgery, and without spending time and research money.

The content lacks the originality of the authors, and the conclusions are nothing new that has been obtained from previous reports, so it is meaningless as a clinical paper. 

Author Response

Dear reviewer,

Thank you a lot for your interest in our submitted paper.

Regarding your comments, we do understand your views. There is nothing more satisfying for a researcher than to publish novel data from own research work. We, as researchers, having published original papers in the past, understand that all too well. However, when referring to the chronic subdural hematoma, the issue is not the absence of data. The problem is that in daily neurosurgical practice, the huge volume of data, which is often conflicting, has led to a lack of consensus, a plethora of expert opinions, and treatment variations, which in turn create a huge heterogeneity in research data, creating a vicious cycle. Since cSDH is such a common entity in neurosurgical departments around the world, every researcher is presenting his or her own case series or comparative studies, which are often not well designed and unable to answer the questions that really matter to neurosurgeons around the world. In the end, the problem is too much data noise. Our work attempts to isolate the signal in all that noise; to distill what we know so far, to make sense of all the data. Original research papers are the crown jewel of every researcher´s curriculum. We, however, focused this time on the clinician. 

We hope our comment helps in seeing the purpose of our paper.

Kind Regards

Mary Solou

Reviewer 2 Report

Dear Authors ,

Thank you r for the extensive review on the subject. Consider to discuss the Endovascular embolization of the middle meningeal artery as an option for specific indications as treatment of SDH.

Best regards,

CB

Author Response

Dear reviewer,

Thank you a lot for your interest in our submitted paper and your recommendations.

Regarding your comment:

“Consider to discuss the Endovascular embolization of the middle meningeal artery as an option for specific indications as treatment of SDH.”

MMA embolization in cSDH seems to be a promising technique, especially in refractory cases. Its use and limitations are under investigation. This is the reason why there are no standard indications for its use, as we have mentioned in our manuscript. Nonetheless, we did include a recent article on page 14, part 4.10 (Nia et al., 2022, Trends and Outcomes of Primary, Rescue, and Adjunct Middle Meningeal Artery Embolization for Chronic Subdural Hematomas, World Neurosurgery), which focuses on indications for primary MMA embolization in certain cases of cSDH.

Hope our corrections fulfill your expectations.

Kind Regards

Mary Solou

Reviewer 3 Report

Reviewer Comments to Author 

Comments to Authors:
Manuscript ID.: 
diagnostics-1842559

Title: T
en Controversies in the Surgical Treatment of Chronic Subdural 
Hematoma: A Systematic Review

Dear Authors,
Overview and general recommendation:
I read your research manuscript with great interest. I think it is important to clarify the clinical evidence of how to manage of CSDH. Many neurosurgeons have been discussing surgical methods for CSDH, but as this study shows, there is no consensus, and methods vary from institution to institution and surgeon to surgeon. Considering this, one problem is that when complications arise, there are no criteria to determine whether or not the indication or method of treatment was appropriate. For neurosurgeons, the lack of consensus on indications and methods of treatment raises the possibility that both patient and surgeon will not reach a satisfactory agreement when a medical accident occurs. Thus, we believe that this study is valuable for social reasons as well.

However, I recommend you had better add one theme to the research, which will reinforce your review. And there is one thing that I think we have to clarify about the research: 

Major comments: 

None.

Minor comments:

1.  Generally, when we speak of treatment of CSDH, we are referring to the supratentorial lesions. However, to date, there are a small number of treatment reports on posterior fossa CSDH. It is not surprising that there are differences in treatment indications and policies between supratentorial and posterior fossa lesions. There are no case series or reviews of posterior fossa CSDH in the text. Therefore, if there is a case series or review on posterior fossa CSDH, it should be mentioned (regardless of the level of evidence, if there is a report, it should be mentioned in the text).

2.  Page 2 of 22, 2. It is stated that The selection of study items followed a double-reviewer process. in the ‘Selection and Data Collection Process’ of Materials and Methods. However, the area of expertise of the reviewers who reviewed the data and papers is unclear. It is presumably M.S. and L.C.S. of Analysist and M.S., L.C.S., I.Y., M.G., E.K.P., S.B., and G.S. of Investigator shown at the end of the text, but it is unclear whether the reviewer of the data/paper was a Radiologist or a Neurosurgeon. I would like to know if the reviewers of the data/paper are Radiologists or a Neurosurgeons, and if they are residents or specialists. It shows the reliability of this research paper. Please clarify.

Author Response

Dear reviewer,

Thank you a lot for your interest in our submitted paper and your recommendations.

Regarding your comment:

“Generally, when we speak of treatment of CSDH, we are referring to the supratentorial lesions. However, to date, there are a small number of treatment reports on posterior fossa CSDH. It is not surprising that there are differences in treatment indications and policies between supratentorial and posterior fossa lesions. There are no case series or reviews of posterior fossa CSDH in the text. Therefore, if there is a case series or review on posterior fossa CSDH, it should be mentioned (regardless of the level of evidence, if there is a report, it should be mentioned in the text).”

Posterior fossa chronic subdural hematomas was a clever observation missing from our manuscript. Indeed, literature has only few case reports regarding this matter. We have adjusted it by adding data about infratentorial hematomas in the section of indications as you recommended: Page 7, paragraph 4.1: “The term cSDH typically refers to supratentorial hematomas. Chronic subdural hematomas of the posterior fossa do occur, they are however rare; in adults they are the result of trauma or other spontaneous causes [26–29]. Apart from case reports and small case series literature is scarce, making formulation of treatment recommendations difficult. Although a few reports on conservative management of cSDH in the posterior fossa do exist, the majority of these hematomas have been treated surgically with either craniotomy, craniectomy, or burr holes [27,29].”.

Regarding your comment:

“Page 2 of 22, 2. It is stated that [The selection of study items followed a double-reviewer process.] in the ‘Selection and Data Collection Process’ of Materials and Methods. However, the area of expertise of the reviewers who reviewed the data and papers is unclear. It is presumably M.S. and L.C.S. of Analysist and M.S., L.C.S., I.Y., M.G., E.K.P., S.B., and G.S. of Investigator shown at the end of the text, but it is unclear whether the reviewer of the data/paper was a Radiologist or a Neurosurgeon. I would like to know if the reviewers of the data/paper are Radiologists or a Neurosurgeons, and if they are residents or specialists. It shows the reliability of this research paper. Please clarify.”

By the word “investigation”, we meant the research around this subject that we did to specify the key points of our manuscript. The reviewing process was the responsibility of M.S. and L.C.S.. We have clarified this in the section of author contributions. M.S. is a senior neurosurgical resident, while L.C.S.  is an Professor of Neurosurgery. A clarification was also inserted in page 2, under “Selection and Data Collection Process”.

Hope our corrections fulfill your expectations.

Kind Regards

Mary Solou